# A structural model of the active ribosome-bound membrane protein insertase YidC

**Stephan Wickles[1,2], Abhishek Singharoy[3], Jessica Andreani[1,2], Stefan Seemayer[1,2], Lukas Bischoff[1,2], Otto Berninghausen[1,2], Johannes Soeding[1,2], Klaus Schulten[3], Eli O van der Sluis[1,2]\*, Roland Beckmann[1,2]\***

[1]Gene Center Munich, Department of Biochemistry, Ludwig-Maximilians-Universität München, Munich, Germany; [2]Center for Integrated Protein Science Munich, Department of Biochemistry, Ludwig-Maximilians-Universität München, Munich, Germany; [3]Beckman Institute for Advanced Science and Technology, University of Illinois at Urbana-Champaign, Urbana, United States

**Abstract** The integration of most membrane proteins into the cytoplasmic membrane of bacteria occurs co-translationally. The universally conserved YidC protein mediates this process either individually as a membrane protein insertase, or in concert with the SecY complex. Here, we present a structural model of YidC based on evolutionary co-variation analysis, lipid-versus-protein-exposure and molecular dynamics simulations. The model suggests a distinctive arrangement of the conserved five transmembrane domains and a helical hairpin between transmembrane segment 2 (TM2) and TM3 on the cytoplasmic membrane surface. The model was used for docking into a cryo-electron microscopy reconstruction of a translating YidC-ribosome complex carrying the YidC substrate $F_O$c. This structure reveals how a single copy of YidC interacts with the ribosome at the ribosomal tunnel exit and identifies a site for membrane protein insertion at the YidC protein-lipid interface. Together, these data suggest a mechanism for the co-translational mode of YidC-mediated membrane protein insertion.

**\*For correspondence:**
vandersluis@lmb.uni-muenchen.
de (EOS); beckmann@lmb.
uni-muenchen.de (RB)

**Competing interests:** The authors declare that no competing interests exist.

**Reviewing editor**: Ramanujan S Hegde, MRC Laboratory of Molecular Biology, United Kingdom

## Introduction

At present, a mechanistic understanding of the function of YidC, as well as its mitochondrial and chloroplast counterparts Oxa1 and Alb3, respectively, is limited by a lack of structural information (*Kol et al., 2008*; *Dalbey et al., 2011*). High resolution structures are available only for the first periplasmic domain (P1) of *Escherichia coli* YidC (*Figure 1A*; *Oliver and Paetzel, 2008*; *Ravaud et al., 2008*), however, this domain is poorly conserved, only present in Gram-negative bacteria and not essential for functionality (*Jiang et al., 2003*). Furthermore, the region(s) of YidC mediating the interaction with the ribosome have not been identified, and the oligomeric state of YidC during co-translational translocation remains controversial (*Kohler et al., 2009*; *Herrmann, 2013*; *Kedrov et al., 2013*). Hence, we set out to determine a molecular model of ribosome-bound YidC during co-translational translocation of the substrate $F_O$c (*van der Laan et al., 2004*), an integral membrane subunit of the ATP synthase complex.

## Results

In order to build an initial structural model of YidC, we predicted contacts between pairs of residues based on covariation analysis (*Marks et al., 2011*; *Hopf et al., 2012*). For that purpose, we constructed a multiple sequence alignment of *E. coli* YidC excluding the nonconserved first transmembrane helix

**eLife digest** Cells are surrounded by a plasma membrane that acts like a barrier to help to keep the cell intact. Proteins are embedded in this plasma membrane; and some of these membrane proteins act as channels that allow molecules to enter and leave the cell, while others allow the cell to communicate with its surroundings.

Like all proteins, membrane proteins are chains of amino acids that are joined together by a molecular machine called a ribosome. Most membrane proteins are inserted into the membrane as they are being built. All bacteria contain a protein called YidC that inserts proteins into the plasma membrane of bacterial cells. However, the mechanism behind this activity and the parts of the YidC protein that interact with the ribosome and plasma membrane are unknown.

Wickles et al. have now used data from a range of sources to predict the three-dimensional structure of the YidC protein taken from a bacterium called *E. coli*. The model shows how the YidC protein is threaded back-and-forth through the membrane, a total of five times. Some of the protein also extends into the inside of the bacterial cell. Wickles et al. then used a technique called cyro-electron microscopy to look at the structure of a YidC protein bound to a ribosome that is building a new protein. Fitting the more detailed model of YidC into this overall structure of the whole complex revealed how a single YidC protein might interact with the ribosome to insert a newly built protein into a membrane.

Wickles et al. then used a combination of theoretical modeling and other experiments to identify the amino acids in the YidC protein that bind to the ribosome: as expected, the binding takes place where the newly formed protein chain exits the ribosome. Further experiments also identified the amino acids in the YidC protein that interact with the newly built membrane protein, thus revealing where it might leave the YidC protein and be inserted into the membrane. The next challenge will be to investigate how the YidC protein assists the folding of new membrane proteins into their own highly specific three-dimensional structure.

(TM1) and the P1 domain (*Figure 1A*) and computed direct evolutionary couplings between pairs of YidC residues (*Kamisetty et al., 2013*). The resulting matrix of coupling strengths (*Figure 1B*) contains several diagonal and anti-diagonal patterns of stronger coupling coefficients, which are indicative of parallel or anti-parallel helix–helix pairs, respectively. We computed probabilities for each possible helix–helix contact by aggregating the evidence of stronger coupling coefficients over the expected interaction patterns and calibrating the resulting raw scores on an independent dataset of helix–helix interactions to obtain accurate interaction probabilities. Seven helix–helix contacts attained probabilities above 57% (*Figure 1B–D*) while all other possible contacts scored below 15%, demonstrating the specificity of the method (*Figure 1—figure supplement 1B*).

We roughly positioned the five TM helices of *E. coli* YidC relative to each other using the predicted helix–helix contacts as constraints, and rotated them according to their predicted lipid or protein exposure (*Lai et al., 2013*; *Figure 1C*). Next, we used MODELLER (*Eswar et al., 2008*) to create full length models based on the TM core, secondary structure prediction and the 50 residue–residue contacts with the highest coupling coefficients (39 excluding intrahelical contacts, indels and topology violations). In the resulting model (*Figure 1E,F*), the conserved membrane integrated core of YidC forms a helical bundle arranged like the vertices of a pentagon, in the order 4-5-3-2-6 (clockwise) when viewed from the cytoplasm (*Figure 1F*). Notably, all the predicted interactions between TM domains can be explained by monomeric YidC suggesting that dimer or oligomer formation may not be strictly required for YidC activity (see also below).

Outside the membrane region, strong helix–helix contacts were predicted within the cytoplasmic loop between TM2 and TM3, which can be explained the by formation of a helical hairpin (*Figure 1F*). The base of this 'helical paddle domain' (HPD) is structurally constrained by predicted contacts with TM3, its tip on the other hand is more mobile and appears to interact with lipid headgroups (see below).

While this manuscript was under review, two crystal structures were published of *Bacillus halodurans* YidC2 (BhYidC2, 34% sequence identity with *E. coli* YidC) (*Kumazaki et al., 2014*), providing us with

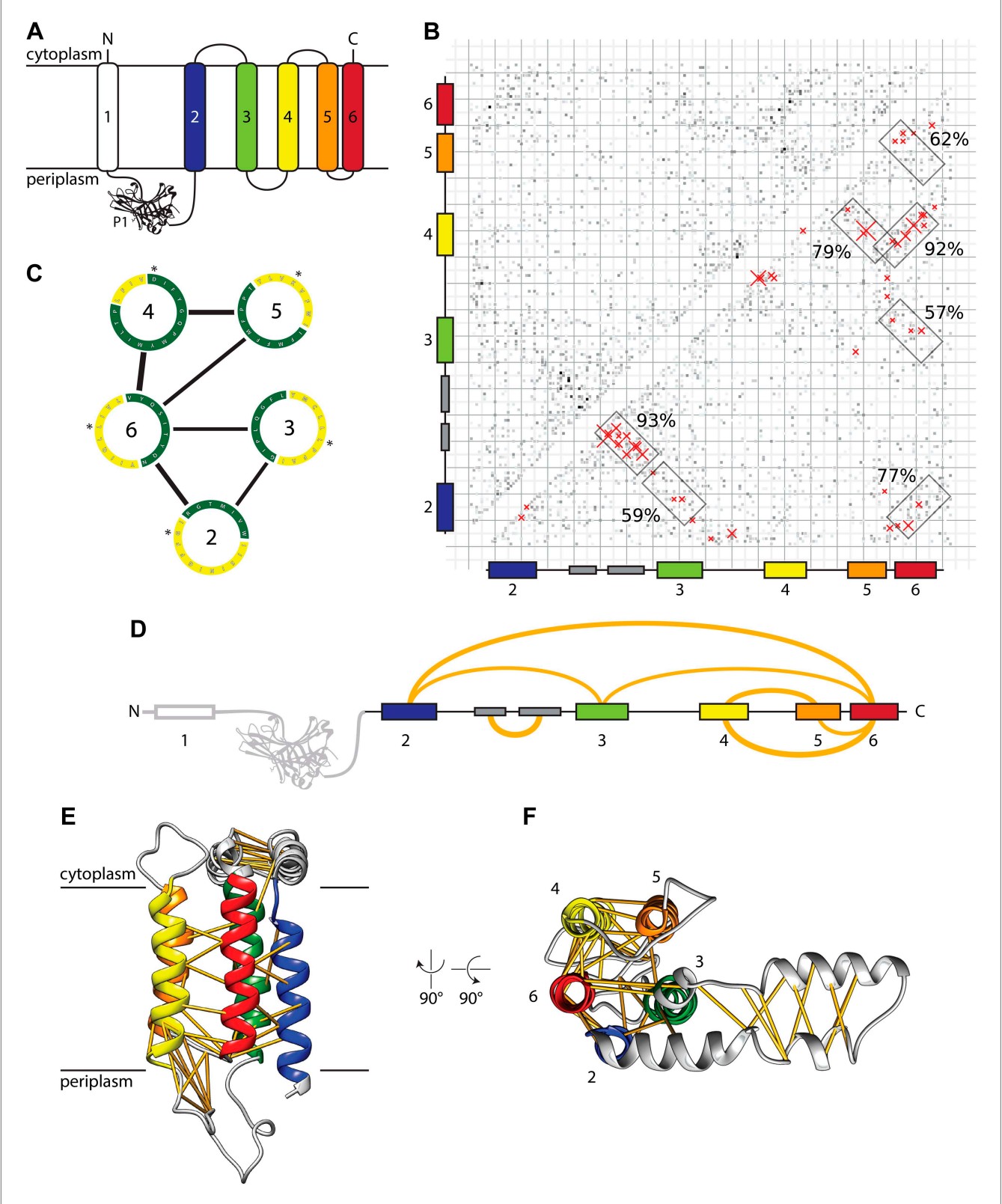

**Figure 1**. Evolutionary covariation based structural model of *E. coli* YidC. (**A**) Membrane topology of YidC, with helix coloring as in all subsequent Figures. (**B**) Matrix of coupling strengths between pairs of YidC residues based on an alignment of 2366 non-redundant sequences. Helix–helix pairs with posterior probabilities higher than 57% are outlined in boxes; the 50 residue–residue pairs with highest coupling coefficients are indicated with red crosses.
*Figure 1. Continued on next page*

*Figure 1. Continued*

(**C**) Overall arrangement of TM helices viewed from the cytoplasm based on the prediction of helix–helix pairs (black lines) and exposure to lipid (yellow) or protein (green). The first residue of each helix is indicated with an asterisk. (**D**) Linear representation of YidC with the seven most probable helix–helix pairs indicated by arches, with thicknesses approximating posterior probabilities. (**E** and **F**) Side view and cytoplasmic view, respectively, of the *E. coli* YidC model based on covariation analysis, with predicted residue–residue pairs indicated by yellow pseudobonds.

The following figure supplements are available for figure 1:

**Figure supplement 1**. Evaluation of possible helix-helix contacts.

a unique opportunity to directly assess the accuracy of our model. Overall, the root mean square deviation (RMSD) between the TM helices of our model and those of BhYidC2 is 7.5 Å (3WO6) and 7.3 Å (3WO7) (*Table 1*), which is within the resolution limits of our method. The global arrangement of TM helices is the same as in BhYidC2, yet, their tilt angle relative to the plane of the membrane is slightly different (*Figure 2*). The tilt angle of the HPD also differs, as well as its side that faces the membrane (*Video 1*), which may be indicative of a high degree of flexibility of this domain, consistent with its high crystallographic B-factors (*Kumazaki et al., 2014*). Notably, the HPD is not essential for YidC function in *E. coli* since the deletion of the entire domain is possible without compromising cell viability (*Jiang et al., 2003*).

A qualitative difference between our model and BhYidC2 that may have more mechanistic importance is the relative position of TM3. In the structure of BhYidC2 a hydrophilic groove is formed on the cytoplasmic side of the TM bundle that has been proposed to form a binding site for YidC substrates (*Kumazaki et al., 2014*). Interestingly, the opening state of this groove differs between the two crystal forms, that is it is more open in 3WO6 than in 3WO7 (*Video 1*), largely due to movement of the N-terminal half of TM3 (*Figure 2—figure supplement 1*). In our model on the other hand, this hydrophilic groove is even more closed than in 3WO7 because we imposed covariation-based constraints between TM3 and TM5 ($Pro^{425}$-$Pro^{499}$) and between TM3 and TM6 ($Cys^{423}$-$Gln^{528}$ & $Phe^{433}$-$Thr^{524}$) (*Figure 2*; *Video 1*). Strikingly, in BhYidC2 the distances between the Cβ atoms of these three pairs are outliers compared to other residue–residue pairs (20.5 Å/20.9 Å/14.9 Å vs an average of 8.2 Å, *Table 2*). Thus, given that (i) the position of TM3 differs in the two crystal forms, and (ii) that covariation analysis predicts with high accuracy a closer interaction of TM3 with TM6 and one contact with TM5, we conclude that movement of TM3 is a genuine feature of YidC. This movement and the accompanying dynamics of the hydrophilic groove may represent a crucial step in the functional cycle of the YidC insertase.

In summary, the overall structure of our YidC model agrees well with the BhYidC2 crystal structure, and a comparison of both structures reveals dynamic regions in YidC that may be of mechanistic importance. This further illustrates the power of covariation analysis not merely for structure prediction but also for obtaining dynamic insights (*Hopf et al., 2012*).

Next, in order to further characterize and validate our obtained YidC model, we assessed its stability and biochemical properties in the bacterial membrane by employing traditional molecular dynamics (MD) simulations. Overall, the model was found to be very stable during the simulation. While the five TM helices enable a rigid protein core, the polar loop regions tend to swim on the membrane surface (*Figure 3A*). An analysis of inter-residue interactions within the TM region (*Figure 3B*) provides a firm basis to the observed stability of YidC: hydrophobic residues on the exterior of the TM bundle stabilize interactions with the apolar lipid tails. The YidC core, in turn, is stabilized both via short and long-range interactions between the five helices. Residues towards the cytoplasmic side of the core are primarily polar or charged and, therefore, engaged in strong electrostatic or charge–dipole interactions. In contrast, residues on the periplasmic side are primarily aromatic and involved in stacking and other nonpolar dispersion interactions.

In order to verify the functional relevance of residues suggested by the MD simulations, we created

**Table 1.** Deviations among YidC structures

|  |  | RMSD (Å) | RMSD (Å) (TM core) |
|---|---|---|---|
| 3WO6 | 3WO7 | 3.1 | 1.8 |
| 3WO6 | model | 9.4 | 7.5 |
| 3WO7 | model | 9.8 | 7.3 |

Overall root mean square deviations (RMSD) between (the TM helices of) our model of *E. coli* YidC and the two BhYidC2 crystal forms.

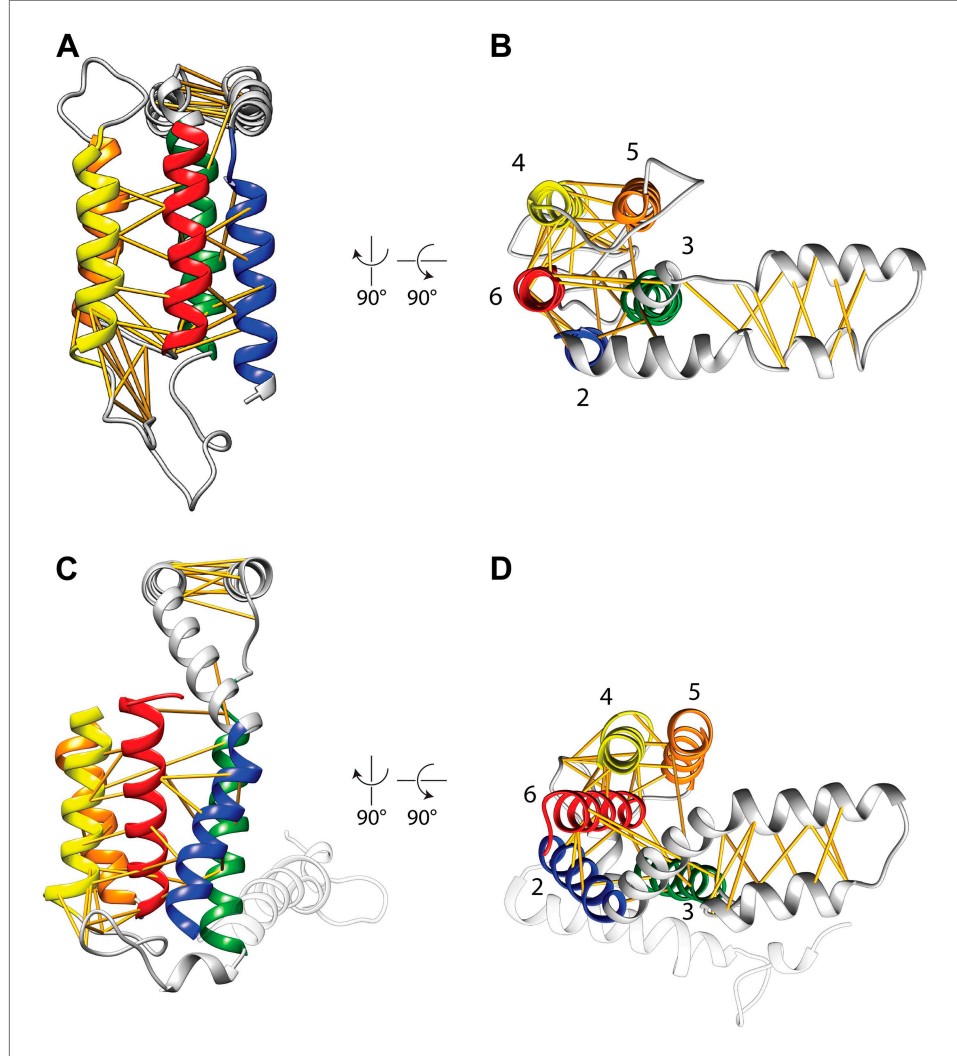

**Figure 2**. Covariation-based model vs homology model. Comparison of the *E. coli* YidC covariation-based model (**A** and **B**) to a homology model of *E. coli* YidC based on the crystal structure of BhYidC2 (3WO6) (**C** and **D**). Predicted residue–residue pairs are indicated by yellow pseudobonds. Note that extracellular helix 1 (white) was not present in our multiple sequence alignment and is thus not included in the model.

The following figure supplements are available for figure 2:

**Figure supplement 1**. Local deviations among YidC structures.

alanine mutants and subjected them to an *in vivo* complementation assay. Some of the most stabilizing residues, T362 in TM2 and Y517 in TM6, both of which are located at the same height in the membrane, completely inactivated YidC when mutated to alanine (*Figure 3D*, *Figure 3—figure supplement 1*). Both mutants were stably expressed, indicating that the lack of complementation was not caused by instability of YidC (*Figure 3—figure supplement 2*). Several residues close to this pair show intermediate activity levels (F433, M471 and F505), whereas residues further away do not show an effect (*Figure 3—figure supplement 1*). Taken together, we provide a model for the overall arrangement of the conserved domains of YidC that is in good agreement with our covariation analysis, lipid exposure prediction, MD simulation, *in vivo* complementation analysis as well as the recent crystal structures.

Interestingly, we observed that YidC induces thinning of the lipid bilayer during the MD simulation. A significant thinning of 7–10 Å results from the hydrophobic mismatch between the TM helices and the membrane (*Figure 3E*). The thinning is similar in the upper and lower leaflet, and the thinnest region is in proximity of TM3 and TM5. Since membrane inserting YidC substrates have been chemically

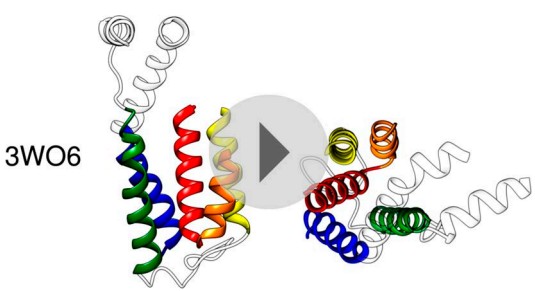

**Video 1**. Conformational states of YidC. Animation showing conformational differences in YidC starting from BhYidC2 crystal form 1 (3WO6), towards crystal form 2 (3WO7) and ending with our covariation based YidC model. Views are from within the membrane (left) and from the cytoplasm (right). Note the movement of the HPD and the closing of the hydrophilic groove between TM3 (orange) and TM5 (green).

cross-linked to both these helices (*Klenner et al., 2008*; *Yu et al., 2008*; *Klenner and Kuhn, 2012*), we argue that thinning of this region in particular may be relevant for the molecular mechanism of YidC-dependent membrane insertion. In addition, the distribution of hydrophilic and hydrophobic residues within YidC revealed the presence of a hydrophilic environment on the cytoplasmic side of the YidC TM bundle (*Figure 3F*), which continues into the mentioned hydrophobic cluster of aromatic residues towards the periplasmic side. It is tempting to speculate that this hydrophilic environment may receive the polar termini and loops of YidC substrates during the initiation of translocation, thus facilitating their transfer across the hydrophobic core of the (thinned) lipid bilayer (see below). Notably, essentially the same conclusions have been drawn on the basis of the BhYidC2 crystal structures and accompanying cross-linking studies (*Kumazaki et al., 2014*).

In order to provide a molecular model of YidC in its active state, we reconstituted purified full length YidC (extended with the C-terminus of *R. baltica* YidC [*Seitl et al., 2014*]) with ribosome nascent chains (RNCs) exposing the first TM helix of $F_Oc$, and subjected the complex to cryo-EM and single particle analysis to a resolution of ~8 Å (*Figure 4A,B*). In agreement with previous structural studies (*Kohler et al., 2009*; *Seitl et al., 2014*), YidC binds to the ribosomal exit site, however, the improved resolution now allows for a more detailed interpretation. Firstly, we were able to separate the weaker electron density of the detergent micelle from that of YidC (*Figure 4A*). Secondly, the presence of elongated structural features (*Figure 4D–F*) allowed us to dock our molecular model in a distinct orientation (cross correlation coefficient 0.865). Following placement of the YidC-core model, two prominent densities in the membrane region, one next to TM3 and one next to TM5, remained unaccounted for. These could be attributed to either TM1 of YidC or to the TM helix of the nascent chain (NC) $F_Oc$. Given that (i) YidC substrates are known to crosslink to TM3 (*Klenner et al., 2008*; *Yu et al., 2008*; *Klenner and Kuhn, 2012*), and (ii) that the density neighboring TM3 is aligned with the ribosomal exit tunnel and (iii) that at the same relative position nascent chains have been observed inside the SecY channel (*Frauenfeld et al., 2011*) (*Figure 4—figure supplement 1*), the most plausible assignment to the density near TM3 appeared to be the TM helix of $F_Oc$. To verify this, and to exclude that the density neighboring TM5 corresponds to the nascent chain, we reconstituted single cysteine mutants of YidC either in TM3 (M430C and P431C) or in TM5 (V500C and T503C) with RNCs of a single cysteine mutant of $F_Oc^{(G23C)}$, and exposed them to disulphide crosslinking. Upon exposure to the oxidator 5,5'-dithiobis-(2-nitrobenzoicacid) (DTNB), only in the TM3 mutants a DTT-sensitive ~90 kDa product appeared that reacted with antibodies against the nascent chain (NC-tRNA~30 kDa, *Figure 4C*) as well as YidC (~60 kDa, *Figure 4C*). Thus, the adduct represented indeed the inserting $F_Oc$ TM domain crosslinked to TM3 of YidC. RNCs lacking a cysteine in the nascent chain (*Figure 4—figure supplement 2*) or YidC mutants with cysteines in TM5 did not yield any crosslinks (*Figure 4C*). Hence, we conclude that the unaccounted electron density next to TM3 represents the TM of the nascent chain, and that the density neighboring TM5 represents TM1 of YidC (*Figure 4D–F*).

We attribute the remaining unaccounted electron density in the periplasmic region to the P1 domain; however, because it is substantially smaller than the crystal structure of P1, we did not include it in our molecular model. Flexibility relative to the conserved membrane region of YidC is the most likely explanation for this finding. We did not observe density for the HPD, in agreement with its flexibility observed in both, the crystal structures of BhYidC2 and the MD simulations (*Figure 3C*).

In order to validate our molecular model of co-translationally active YidC, we mutated residues that would be in direct contact with the ribosome (*Figure 5A,B*) and analyzed their effect on functionality in the *in vivo* complementation test. Indeed, mutation of residues Y370A and Y377A (contacting ribosomal RNA helix 59) and D488K (contacting ribosomal protein uL23) severely interfere with YidC activity (*Figure 5C*, *Figure 5—figure supplement 1*) thereby emphasizing their functional significance.

**Table 2.** Top 50 scoring residue–residue pairs in covariation analysis

| Residue 1 | # Residue 1 | Region | | Residue 2 | # Residue 2 | Region | dmodel (Å) | d3WO6 (Å) | Reason for exclusion |
|---|---|---|---|---|---|---|---|---|---|
| TRP | 354 | TM2 | | – | – | – | | | indel |
| GLY | 355 | TM2 | | – | – | – | | | indel |
| PHE | 356 | TM2 | | – | – | – | | | indel |
| PHE | 356 | TM2 | | ARG | 533 | c-term | | | topology violation |
| ILE | 358 | TM2 | <–> | GLY | 512 | Loop5-6 | 9.1 | 6.1 | |
| ILE | 359 | TM2 | <–> | VAL | 519 | TM6 | 6.5 | 5.2 | |
| ILE | 359 | TM2 | <–> | LEU | 515 | TM6 | 8.5 | 7.9 | |
| ILE | 359 | TM2 | | – | – | – | | | indel |
| ILE | 361 | TM2 | <–> | LEU | 436 | TM3 | 7.9 | 8.2 | |
| THR | 362 | TM2 | | PRO | 371 | TM2 | | | intrahelical |
| PHE | 363 | TM2 | <–> | VAL | 523 | TM6 | 5.2 | 6.1 | |
| GLY | 367 | TM2 | <–> | VAL | 523 | TM6 | 6.0 | 8.2 | |
| MET | 369 | TM2 | <–> | ILE | 432 | TM3 | 9.9 | 8.4 | |
| Leu | 372 | Loop2-3 | | PRO | 510 | Loop5-6 | | | topology violation |
| SER | 379 | Loop2-3 | <–> | PRO | 425 | TM3 | 10.2 | 9.9 | |
| LEU | 386 | Loop2-3 | <–> | VAL | 417 | Loop2-3 | 7.5 | 7.1 | |
| LEU | 386 | Loop2-3 | <–> | LEU | 411 | Loop2-3 | 6.2 | 6.1 | |
| PRO | 388 | Loop2-3 | | GLN | 429 | TM3 | | | topology violation |
| LYS | 389 | Loop2-3 | <–> | ALA | 414 | Loop2-3 | 10.5 | 9.8 | |
| LYS | 389 | Loop2-3 | <–> | GLU | 415 | Loop2-3 | 11.2 | 10.0 | |
| ILE | 390 | Loop2-3 | <–> | MET | 408 | Loop2-3 | 6.8 | 6.2 | |
| MET | 393 | Loop2-3 | <–> | ILE | 404 | Loop2-3 | 7.9 | 7.4 | |
| MET | 393 | Loop2-3 | <–> | LEU | 411 | Loop2-3 | 8.2 | 7.7 | |
| ARG | 394 | Loop2-3 | <–> | ILE | 404 | Loop2-3 | 8.5 | 8.1 | |
| ARG | 396 | Loop2-3 | <–> | GLU | 407 | Loop2-3 | 8.9 | 8.4 | |
| CYS | 423 | TM3 | <–> | GLN | 528 | TM6 | 16.2 | 20.9 | |
| PRO | 425 | TM3 | <–> | PRO | 499 | TM5 | 10.2 | 20.5 | |
| PHE | 433 | TM3 | <–> | THR | 524 | TM6 | 11.0 | 14.9 | |
| LEU | 436 | TM3 | <–> | GLY | 512 | Loop5-6 | 7.6 | 8.3 | |
| TYR | 437 | TM3 | <–> | LEU | 513 | Loop5-6 | 9.8 | 6.4 | |
| TRP | 454 | Loop3-4 | <–> | ASP | 462 | Loop3-4 | 6.6 | 7.0 | |
| TRP | 454 | Loop3-4 | <–> | PRO | 468 | TM4 | 16.0 | 11.5 | |
| TRP | 454 | Loop3-4 | <–> | SER | 511 | Loop5-6 | 9.8 | 8.3 | |
| ILE | 455 | Loop3-4 | <–> | LEU | 467 | TM4 | 9.8 | 10.1 | |
| ILE | 455 | Loop3-4 | <–> | ILE | 466 | TM4 | 11.0 | 8.0 | |
| ASP | 462 | Loop3-4 | <–> | PRO | 468 | TM4 | 12.5 | 6.8 | |
| ASP | 462 | Loop3-4 | <–> | SER | 511 | Loop5-6 | 11.1 | 4.2 | |
| TYR | 465 | TM4 | <–> | LEU | 507 | TM5 | 10.4 | 8.7 | |
| LEU | 467 | TM4 | <–> | LEU | 515 | TM6 | 11.6 | 6.6 | |
| PRO | 468 | TM4 | <–> | LEU | 513 | TM6 | 14.5 | 8.8 | |
| LEU | 470 | TM4 | <–> | ILE | 518 | TM6 | 6.3 | 5.4 | |
| MET | 471 | TM4 | <–> | PHE | 502 | TM5 | 8.8 | 4.9 | |
| GLY | 472 | TM4 | <–> | THR | 503 | TM5 | 6.7 | 5.3 | |
| GLY | 472 | TM4 | | GLN | 479 | TM4 | | | intrahelical |

*Table 2. Continued on next page*

*Table 2. Continued*

| Residue 1 | # Residue 1 | Region | | Residue 2 | # Residue 2 | Region | dmodel (Å) | d3WO6 (Å) | Reason for exclusion |
|---|---|---|---|---|---|---|---|---|---|
| THR | 474 | TM4 | <–> | ASN | 521 | TM6 | 4.7 | 3.7 | |
| THR | 474 | TM4 | <–> | ILE | 525 | TM6 | 6.7 | 7.8 | |
| ILE | 478 | TM4 | <–> | ILE | 525 | TM6 | 9.0 | 5.0 | |
| *THR* | *485* | *Loop4-5* | | *–* | *–* | *–* | | | *indel* |
| PHE | 506 | TM5 | <–> | VAL | 514 | TM6 | 14.4 | 4.2 | |
| *GLY* | *512* | *Loop5-6* | | *GLN* | *532* | *TM6* | | | *topology violation* |
| | | | | | | Ø | 9.3 | 8.1 | |

Table showing the 50 residue–residue pairs with the highest covariation scores, and the distances between the Cβ atoms in the final model of the 39 pairs that were used as constraints for model building. For comparison, the corresponding distances in 3WO6 are also given. The 11 residue–residue pairs that were excluded for model building are in italics, with the reason for their exclusion indicated on the right.

All these mutants were stably expressed, indicating that the lack of complementation was not caused by instability of YidC (*Figure 5—figure supplement 2*). Given that YidC in general is known to be very tolerant to point mutations (*Jiang et al., 2003*), this provides further support for the overall correctness of our model of ribosome-bound YidC during membrane protein insertion.

## Discussion

Finally, it is notable that we observe only a single monomer of YidC bound to the active ribosome. This is in agreement with recent literature showing clearly that both YidC (*Herrmann, 2013*; *Kedrov et al., 2013*; *Seitl et al., 2014*) and the SecY complex (*Frauenfeld et al., 2011*; *Park and Rapoport, 2012*; *Taufik et al., 2013*; *Park et al., 2014*) can be fully active as monomers. However, the comparison of models for active YidC and active SecY (*Figure 5E*, *Figure 4—figure supplement 1*) reveals an important difference between the two proteins that has mechanistic implications. While SecY is known to translocate hydrophilic nascent chains through its central aqueous channel (*Cannon et al., 2005*; *Rapoport, 2007*; *Driessen and Nouwen, 2008*) and insert TM domains through a lateral gate (*Van den Berg et al., 2004*; *Gogala et al., 2014*), our model suggests that the YidC substrates are inserted at the protein-lipid interface. Two principal findings of our work suggest how YidC may facilitate this process: (i) it provides a hydrophilic environment within the membrane core for receiving the hydrophilic moieties (termini or loops) of a substrate, and (ii) it reduces the thickness of the lipid bilayer: initial interaction of the hydrophilic moieties of YidC substrates with the hydrophilic environment of YidC would allow for a partial insertion into the membrane, while facilitating exposure of the hydrophobic TM domains to the hydrophobic core of the bilayer. The latter in turn may compensate for the energetic penalty of driving the hydrophilic moieties across the (already thinned) bilayer. Further biochemical and structural studies that capture the earlier stages of this translocation process are eagerly awaited to fully elucidate this mechanism.

## Materials and methods

### Covariation analysis

We constructed a multiple sequence alignment of YidC excluding the unconserved first transmembrane helix (TM1) and the periplasmic P1 domain. We searched for homologous sequences of *E. coli* YidC starting from the PFAM seed alignment of family PF02096 (*Punta et al., 2012*) and using the sensitive homology detection software HHblits (*Remmert et al., 2012*). First, five iterations of HHblits were run against the clustered Uniprot database with no filtering, to retrieve as many homologous sequences as possible. Then, we post-processed the alignment using HHfilter to generate a non-redundant alignment at 90% sequence identity. This resulted in an alignment containing 2366 sequences aligned across YidC helices TM2-TM6. Using this multiple sequence alignment, we computed direct evolutionary couplings between pairs of YidC residues using the method of *Kamisetty et al. (2013)*.

To compute probabilities for each possible helix–helix contact, we aggregated the evidence of stronger coupling coefficients over the expected interaction patterns for helix–helix contacts, taking into account the expected periodicity of ~3.5 residues per alpha helix turn. We built three non-redundant

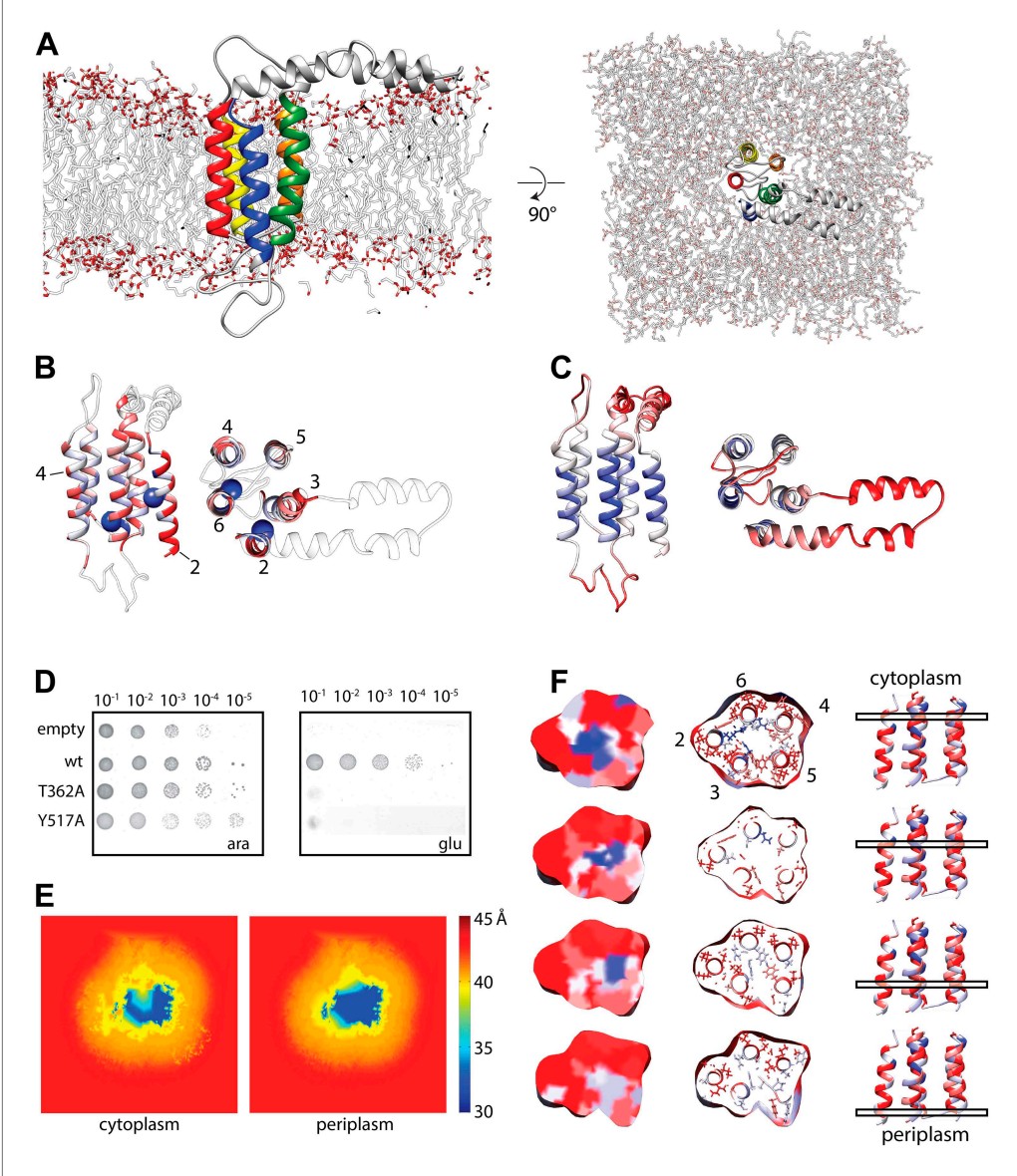

**Figure 3**. Molecular dynamics simulation of the YidC model. (**A**) Side view (left) and cytoplasmic view (right) of the stable YidC model after a 500 ns MD simulation in a lipid bilayer composed of 3:1 POPE:POPG. (**B**) Ribbon representation of the stable model according to inter-helix energy (in kcal/mol), blue: −7.5 to −1; white: −1 to −0.002; red: ≥ −0.00.2. Residues that inactivate YidC upon mutagenesis are indicated by spheres. (**C**) Ribbon representation of the stable model according to flexibility (in Å²), blue: 0.04 to 0.09; white: 0.09–1; red: ≥1.0. (**D**) *In vivo* complementation assay of YidC mutants T362A (TM2) and Y517A (TM6). (**E**) Thickness of the cytoplasmic and periplasmic leaflet of the simulated bilayer after 500 ns, highlighting the membrane thinning effect in the vicinity of YidC. The membrane surface is defined by positions of polar head groups in the lipids, and thickness at a given point on the surface is taken to be the shortest distance between the head groups from opposite leaflets. The thickness values are averaged over the MD trajectory and presented as a contour plot on the membrane surface with a color-scale from red, indicating thicker region representing bulk bilayer lipids, to blue showing thinned regions close to YidC suggesting hydrophobic mismatch. (**F**) Distribution of hydrophobic (red) and hydrophilic residues (blue) in YidC at various heights of the membrane, highlighting the hydrophilic environment in the center of YidC on the cytoplasmic side.

The following figure supplements are available for figure 3:

**Figure supplement 1**. Complementation of MD-based mutants.

**Figure supplement 2**. Expression of MD-based mutants.

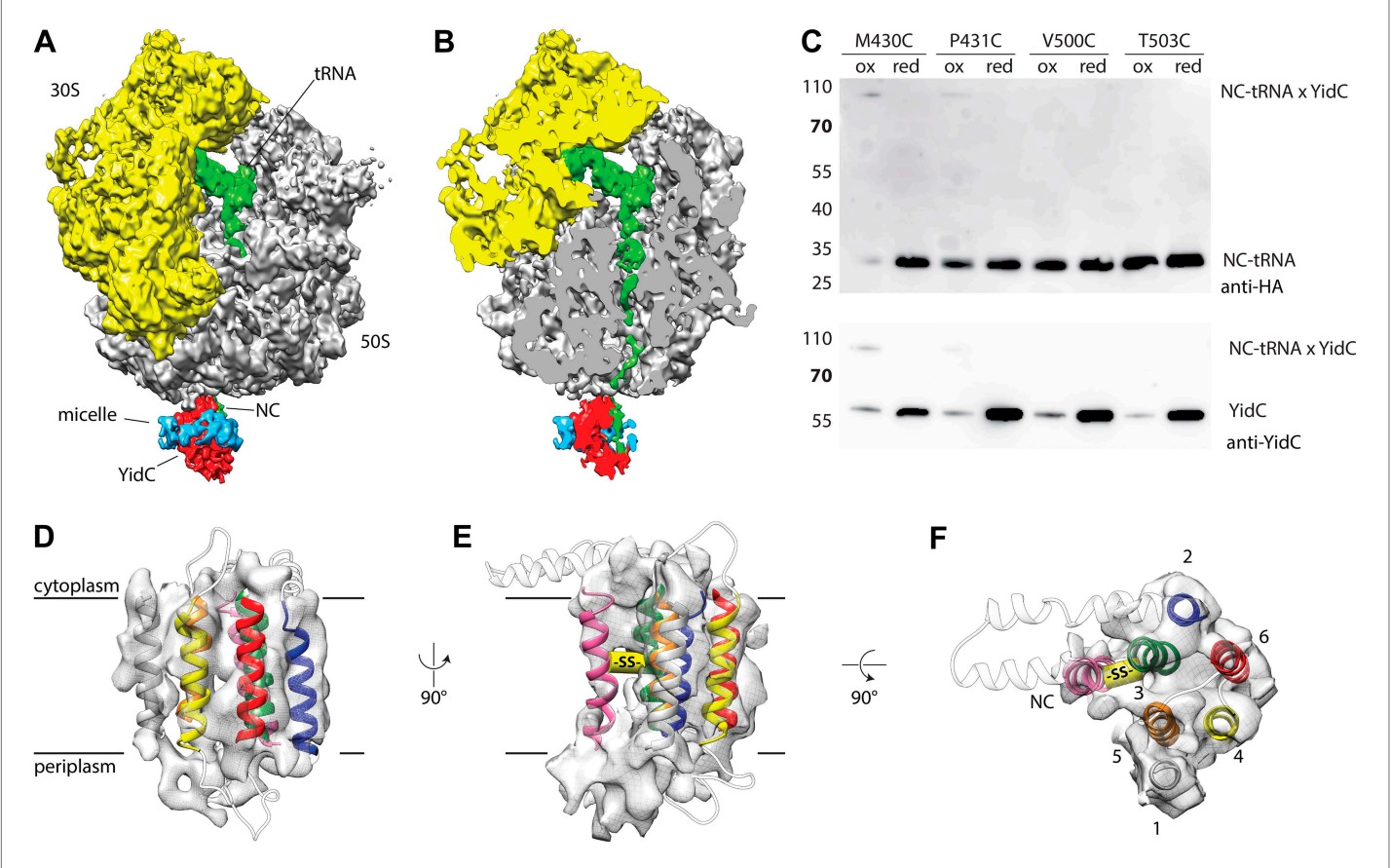

**Figure 4**. Cryo-EM structure of RNC bound YidC and structural model of the active state. (**A**) Side view of the ~8 Å resolution cryo-EM based electron density of the RNC:YidC complex, with the small subunit depicted in yellow, the large subunit in gray, P-site tRNA and nascent chain in green, YidC in red and the detergent micelle in blue. (**B**) As in **A**, but sliced through the ribosomal exit tunnel. (**C**) Validation of the active state model by disulphide crosslinking. RNCs carrying the mutant $F_Oc^{(G23C)}$ were reconstituted with the indicated single cysteine YidC mutants, oxidized, applied to a linear sucrose gradient and harvested from the 70S peak before SDS-PAGE and western blotting. Immunodetection was performed with antibodies raised against the HA-tag (located in the nascent chain inside the ribosomal exit tunnel) and anti-YidC antibodies. YidC, nascent chain-tRNA (NC-tRNA) and the expected crosslink product (NC-tRNA x YidC) are indicated. (**D**–**F**) Structural model of YidC during membrane protein insertion, viewed from two sides within the membrane (**D** and **E**) and from the cytoplasm (**F**). The detergent micelle was removed for clarity, the TM helix of $F_Oc$ is depicted in magenta, and the disulphide crosslink between YidC and $F_Oc$ with -SS-.

The following figure supplements are available for figure 4:

**Figure supplement 1**. Comparison of the active states of YidC and SecY.

**Figure supplement 2**. Negative control for RNC-YidC crosslinking.

datasets of mainly-alpha proteins from the CATH database (*Sillitoe et al., 2013*). For each protein, we slid a square pattern (of size 17 × 17 residues = 289 cells) over the matrix of coupling strengths. For each pattern position, we used Bayes theorem to calculate the raw probability for a helix–helix interaction, given the 289 coupling strengths. The distributions of coupling strengths for interacting and non-interacting helix residues were fitted on dataset #1 (1118 proteins). We assigned different weights to the pattern cells, depending on their position within the pattern and the direction of the helix–helix interaction (parallel or antiparallel); these weights were optimized on dataset #2 (204 proteins). Finally, we calibrated the resulting raw scores on dataset #3 (85 proteins) to obtain accurate interaction probabilities. For cross-validation purposes, we also performed optimization on dataset #3 and calibration on dataset #2. Optimization on either dataset #2 or dataset #3 results in the same choice of weights for the pattern cells. The final posterior probabilities were obtained as the average of the values calibrated on datasets

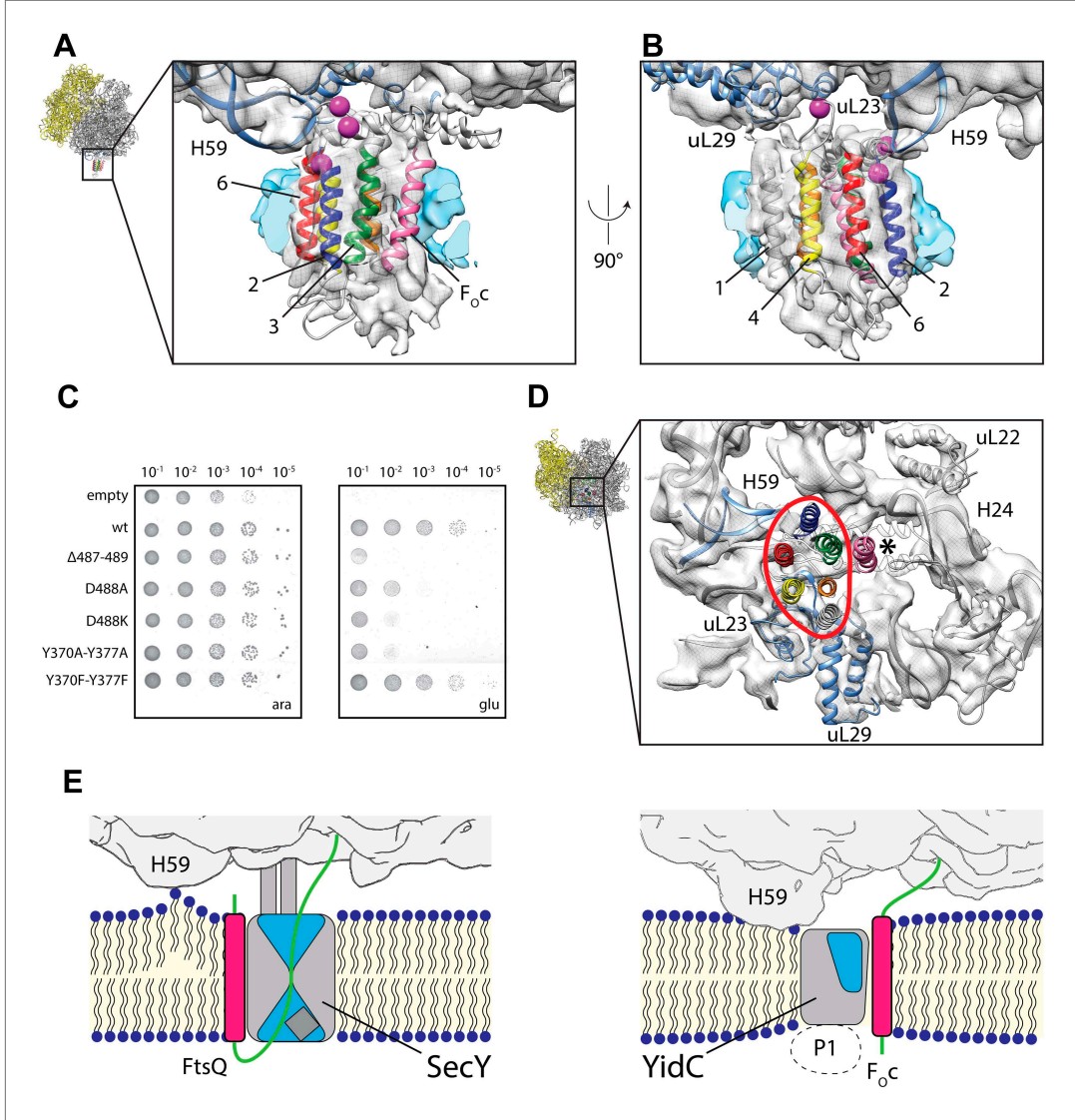

**Figure 5**. Contacts between active YidC and the ribosome. (**A** and **B**) Close-up views from within the membrane region highlighting the contact between H59 of the ribosome and the 2/3 loop of YidC (**A**) and ribosomal protein uL23 and the 4/5 loop of YidC (**B**). Residues that inactivate YidC upon mutagenesis or deletion are indicated by magenta spheres. (**C**) *In vivo* complementation assay of YidC point mutants D488A, D488K, deletion mutant Δ487-489 and the double mutants Y370A/Y377A and Y370F/Y377F. (**D**) Periplasmic view of the active ribosome-bound YidC model, with the YidC contour outlined in red. The polypeptide exit tunnel is indicated with an asterisk. (**E**) Cartoon based comparison of active SecY (left) and active YidC (right) during membrane insertion of FtsQ and $F_Oc$, respectively. The ribosome is depicted in gray, the aqueous channel in SecY as well as the hydrophilic environment within YidC are shaded blue, hydrophobic TM domains of the substrates are depicted magenta, hydrophilic parts in green and the P1 domain by a dashed oval.

The following figure supplements are available for figure 5:

**Figure supplement 1**. Complementation of ribosome interaction mutants.

**Figure supplement 2**. Expression of ribosome interaction mutants.

#2 and #3, weighted by dataset size. The calibration plots for datasets #2 and #3 are shown in *Figure 1—figure supplement 1A*. The histogram of final posterior probabilities obtained for YidC is shown in *Figure 1—figure supplement 1B*, which illustrates the specificity of the helix–helix predictions.

## YidC initial model building

The conserved TM helices of *E.coli* YidC were positioned according to the covariation based helix–helix contact prediction, and rotated based on their predicted lipid or protein exposure (*Lai et al., 2013*), resulting in a starting model of the conserved TM core of YidC. Additional information based on direct residue–residue interactions (covariance analysis) and secondary structure predictions by Jpred 3 (*Cole et al., 2008*) were used as structural restraints in MODELLER (*Eswar et al., 2008*). From a total of 10 output models that differed mainly in the relative orientation of the loop regions, the model that satisfied the imposed constraints best was used for further studies.

## Molecular dynamics simulation

### System preparation

All simulations were performed with the MD software NAMD 2.9 using the CHARMM36 force field for the proteins and lipids (*Klauda et al., 2010*). The TIP3P model is used to simulate water (*Jorgensen et al., 1983*). The YidC model was inserted into the membrane, solvated, and ionized using the Membrane Builder tools on CHARMM-GUI (*Jo et al., 2008*). The lipid composition is chosen to be 3 POPE to 1 POPG, as has been successfully used for modeling bacterial membranes in several past MD simulations (*Ash et al., 2004*; *Mondal et al., 2013*). An initial membrane surface of area 110 Å × 110 Å was constructed along the XY plane. The protein lipid-construct was solvated with 25 Å thick layers of water along the Cartesian Z directions, and ionized to charge neutralization using Monte Carlo sampling of $Na^+$ and $Cl^-$ ions at 0.15 M concentration. The overall system size is 0.15 M. Prior to simulation the system was subjected to 10,000 steps of conjugate gradient energy minimization, followed by 100 ps of thermalization and 25 ns of equilibration. During the first 10 ns of the equilibration stage, the protein was kept fixed, allowing the lipids, ions and water molecules to equilibrate. Subsequent 15 ns of equilibration included the protein as well. We then performed 500 ns of MD simulation at 300 K. The final 100 ns was repeated thrice to examine the statistical significance of the result.

### Simulation parameters

The systems were kept at constant temperature using Langevin dynamics for all non-hydrogen atoms with a Langevin damping coefficient of 5 $ps^{-1}$. A constant pressure of 1 atm was maintained using the Nose-Hoover Langevin piston with a period of 100 fs and damping timescale of 50 fs. Simulations were performed with an integration time step of 1 fs where bonded interactions were computed every time step, short-range non-bonded interactions every two time steps, and long range electrostatic interactions every four time steps. A cutoff of 12 Å was used for van der Waals and short-range electrostatic interactions: a switching function was started at 10 Å for van der Waals interactions to ensure a smooth cutoff. The simulations were performed under periodic boundary conditions, with full-system, long-range electrostatics calculated by using the PME method with a grid point density of 1/Å. The unit cell was large enough so that adjacent copies of the system did not interact via short-range interactions.

### Flexibility analysis

The overall flexibility of the transmembrane helices relative to their average configuration was compared. Positional variance of the helix residues was quantified as a measure of their flexibility. Positional variance was computed by summing the deviation of individual backbone atom position and dividing by the number of backbone atoms in the loop. This measure is slightly different from the usual root mean square fluctuation (RMSF) as contributions from overall displacements of the helices and their motions relative to the rotation/translation and internal motions of the protein are included to probe flexibility.

### Interaction energy, hydrogen bonds, and membrane thickness analysis

To further understand the details of the structure and dynamics of the YidC model we performed interaction energy, hydrogen bond, and membrane thinning analysis. These analyses were carried out on the MD trajectory using standard tools available on VMD. In particular, interaction energies were computed for each trajectory frame of the final 100 ns simulation using the NAMD Energy plugin on VMD. The numbers were then time averaged over the entire 100 ns, locally averaged for every residue over a cut-off distance of 10 Å, and plotted on the structure in *Figure 3B*. Hydrogen bonds are defined

solely on the basis of geometric parameters (bond angle: 20°; bond-length: 3.8 Å) between donors and acceptors. Thickness at a given point on the membrane surface was probed by finding the nearest lipid head group and measuring the minimum distance between the phosphate on that lipid head and one on the opposite leaflet.

## Purification of ribosome nascent chain complexes (RNCs)

RNC constructs encoding residues 1–46 of $F_Oc$ (preceded by an N-terminal His-tag and 3C rhinoprotease cleavage site, and followed by an HA-tag and TnaC stalling sequence) were cloned into a pBAD vector (Invitrogen, Life Technologies, Karlsruhe, Germany) by standard molecular biology techniques, and expressed and purified as described before (*Bischoff et al., 2014*). Briefly, *E.coli* KC6 Δ*smpB*Δ*ssrA* (*Seidelt et al., 2009*) carrying the plasmid for $F_Oc$ was grown in LB with 100 µg/ml ampicilin at 37°C to an $OD_{600}$ = 0.5 and expression was induced for 1 hr by adding 0.2% arabinose. Cells were lysed and debris was removed by centrifugation for 20 min at 16.000 rpm in a SS34-rotor (Sorvall). The cleared lysate was spun overnight through a sucrose cushion at 45.000 rpm in a Ti45 rotor (Beckmann), the ribosomal pellet was resuspended for 1 hr at 4°C and RNCs were purified in batch by affinity purification using Talon (Clontech). After washing the Talon beads with high salt buffer the RNCs were eluted and loaded onto a linear 10%–40% sucrose gradient. The 70S peak was collected, RNCs were concentrated by pelleting, resuspended in an appropriate volume of RNC Buffer (20 mM HEPES pH 7.2, 100 mM KOAc, 6 mM MgOAc$_2$, 0.05% (wt/vol) dodecyl maltoside), flash frozen in liquid $N_2$ and stored at −80°C. The complete sequence of the nascent chain is:

MGHHHHHHHHDYDIPTTLEVLFQGPGTMENLNMDLLYMAAAVMMGLAAIGAAIGIGILGGKFLEG AARQPDLIYPYDVPDYAGPNILHISVTSKWFNIDNKIVDHRP.

## Purification of YidC

For purification and reconstitution studies, *E.coli* YidC extended with the C-terminus from *R. baltica* (*Seitl et al., 2014*) was re-cloned into pET-16 (Novagen) with an N-terminal His-tag followed by a 3C rhinovirus protease site. Expression and purification was performed essentially as described (*Lotz et al., 2008*). Briefly, *E.coli* C43(DE3) cells (*Miroux and Walker, 1996*) harboring the YidC construct were grown at 37°C to an $OD_{600}$ = 0.6 and expression was induced by adding 0.5 mM IPTG. YidC was solubilized with Cymal-6 (Anatrace) and purified by affinity chromatography using TALON (Clontech). The N-terminal His-tag of the eluted protein was cleaved off with 3C protease during overnight dialysis at 4°C, followed by gel filtration chromatography (Superdex 200; GE Healthcare). Fractions of the monodisperse peak were pooled, concentrated to ~1 mg/ml in YidC Buffer (20 mM NaPO$_4$ pH 6.8, 100 mM KOAc, 10% glycerol, 0.05% Cymal-6) and directly used for further structural or biochemical assays.

## Disulphide crosslinking

For disulphide crosslink analysis, $F_Oc^{(G23C)}$-RNCs and single cysteine mutants of YidC were purified separately and reconstituted by incubating 100 pmol of RNCs with 500 pmol of freshly purified YidC for 30 min at 37°C. The endogenous cysteine in YidC at position 423 was replaced by serine. Disulphide crosslinking was induced by adding 1 mM 5,5′-dithiobis-(2-nitrobenzoicacid) (DTNB) for 10 min at 4°C and quenched by adding 20 mM N-Ethylmaleimide (NEM) for 20 min at 4°C. Crosslinked RNC-YidC complexes were separated from non-crosslinked YidC using a 10%–40% linear sucrose gradient, and the 70S peak was harvested and analyzed by SDS-PAGE followed by western blotting.

## Complementation assay

For *in vivo* complementation studies, wildtype *E. coli* YidC was recloned into pTrc99a (Pharmacia), and mutants were created by standard molecular cloning techniques. *E.coli* FTL10 cells (*Hatzixanthis et al., 2003*) harboring pTrc99a plasmids encoding the YidC variants were grown overnight at 37°C in LB medium supplemented with 100 µg/ml ampiciline, 50 µg/ml kanamycin and 0.2% arabinose. YidC depletion was carried out by transferring the cells to LB medium supplemented with 100 µg/ml ampiciline, 50 µg/ml kanamycin and 0.2% glucose, followed by and additional incubation for 3 hr at 37°C. Cell suspensions of all constructs were adjusted to $OD_{600}$ = 0.1 and either loaded onto SDS-PAGE gels for subsequent Western blotting, or further diluted to $OD_{600}$ = 10$^{-5}$. Each dilution was spotted on LB agar plates supplemented 100 µg/ml ampiciline, 50 µg/ml kanamycin and either 0.2% arabinose or 0.2% glucose, and incubated overnight at 37°C.

## Electron microscopy and image processing

For cryo-EM analysis, $F_O$c-RNC:YidC complexes were reconstituted by incubating 10 pmol of RNCs with 100 pmol of freshly purified YidC for 30 min at 37°C in a final volume of 50 µl of RNC buffer. Samples were applied to carbon-coated holey grids according to standard methods (*Wagenknecht et al., 1988*). Micrographs were collected under low-dose conditions on a FEI TITAN KRIOS operating at 200 kV using a 4 k × 4 k TemCam-F416 CMOS camera and a final pixel size of 1.035 Å on the object scale.

Image processing was done using the SPIDER software package (*Shaikh et al., 2008*). The defocus was determined using the TF ED command in SPIDER followed by automated particle picking using Signature (*Chen and Grigorieff, 2007*). The machine-learning algorithm MAPPOS (*Norousi et al., 2013*) was used to subtract 'false positive' particles from the data set and initial alignment was performed using an empty 70S ribosome as reference. The complete data set (876376 particles) was sorted using competitive projection matching in SPIDER followed by focused sorting for ligand density (*Leidig et al., 2013*), and refined to a final resolution of ~8.0 Å (Fourier shell correlation [FSC] cut-off 0.5). The final dataset consisted of 58,960 particles showing electron density for P-site tRNA and ligand density at the tunnel exit. We have deposited our cryo-EM map at the EMDB under accession number 2705, and the model of the transmembrane domains at the PDB under accession number 4utq.

## Acknowledgements

We would like to thank C Ungewickell for assistance with cryo-electron microscopy, Susan Vorberg for assistance with covariation analyses, T Palmer for providing *E. coli* strain FTL10, A Driessen and A Kuhn for providing YidC antibodies, J Philippou-Massier and U Gaul for use of the robotic high-throughput facility, A Heuer for assistance with animations and B Beckert and A Kedrov for discussions.

SW and LB were supported by the International Max Planck Research School, SS by grant GRK1721 from the DFG, JA by a Humboldt Research Felloship of the Alexander-von-Humboldt Foundation and the Bavarian Network for Molecular Biosystems (BioSysNet), AS by a Beckman Postdoctoral Fellowship, KS by the Center for Macromolecular Modeling and Bioinformatics (NIH 9P41GM104601, NIH R01-GM67887) and the Center for the Physics of Living Cells (NSF PHY-0822613), JS by the Deutsche Forschungsgemeinschaft (DFG) trough grants SFB646, GRK1721, and QBM, by the Bundesministerium für Bildung und Forschung through grant CoreSys and the Bavarian Network for Molecular Biosystems (BioSysNet), and RB by the Center for Integrated Protein Science, the DFG (FOR967) and the European Research Council (Advanced Grant CRYOTRANSLATION).

## Additional information

### Funding

| Funder | Grant reference number | Author |
| --- | --- | --- |
| Deutsche Forschungsgemeinschaft | GRK1721, SFB646, QBM | Johannes Soeding |
| Alexander von Humboldt-Stiftung | Research Fellowship | Jessica Andreani |
| National Institutes of Health | Center for Macromolecular Modeling and Bioinformatics, 9P41GM104601 | Abhishek Singharoy, Klaus Schulten |
| National Science Foundation | Center for the Physics of Living Cells, PHY-0822613 | Abhishek Singharoy, Klaus Schulten |
| European Research Council | CRYOTRANSLATION | Roland Beckmann |
| Max-Planck-Gesellschaft | International Max Planck Research School | Stephan Wickles, Lukas Bischoff |
| Bavarian Network for Molecular Biosystems (BioSysNet) | | Jessica Andreani, Johannes Soeding |
| Beckman Institute for Advanced Science and Technology, University of Illinois, Urbana–Champaign | Postdoctoral Fellowship | Abhishek Singharoy |

| Funder | Grant reference number | Author |
|---|---|---|
| National Institutes of Health | Center for Macromolecular Modeling and Bioinformatics, R01-GM67887 | Klaus Schulten |
| Bundesministerium für Bildung und Forschung | CoreSys | Johannes Soeding |
| Centre for Integrated Protein Science (CIPSM) | | Roland Beckmann |
| Deutsche Forschungsgemeinschaft | FOR967 | Roland Beckmann, Eli O van der Sluis |
| Deutsche Forschungsgemeinschaft | GRK1721 | Stefan Seemayer |

The funders had no role in study design, data collection and interpretation, or the decision to submit the work for publication.

## Author contributions

SW, AS, Conception and design, Acquisition of data, Analysis and interpretation of data, Drafting or revising the article; JA, SS, Acquisition of data, Analysis and interpretation of data, Drafting or revising the article; LB, OB, Acquisition of data, Analysis and interpretation of data; JS, KS, EOS, RB, Conception and design, Analysis and interpretation of data, Drafting or revising the article

# Additional files

## Major dataset

The following datasets were generated:

| Author(s) | Year | Dataset title | Dataset ID and/or URL | Database, license, and accessibility information |
|---|---|---|---|---|
| Wickles S, Singharoy A, Andreani J, Seemayer S, Bischoff L, Berninghausen O, Soeding J, Schulten K, O van der Sluis E, Beckmann R | 2014 | A structural model of the active ribosome-bound membrane protein insertase YidC | 2705; http://www.ebi.ac.uk/pdbe/entry/EMD-2705 | Publicly available at the EMBL-EBI EMDB. |
| Wickles S, Singharoy A, Andreani J, Seemayer S, Bischoff L, Berninghausen O, Soeding J, Schulten K, O van der Sluis E, Beckmann R | 2014 | A structural model of the active ribosome-bound membrane protein insertase YidC | 4utq; http://www.ebi.ac.uk/pdbe-srv/view/entry/4utq/summary.html | Publicly available at the EMBL-EBI Protein Data Bank in Europe. |

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
