## [Decision Letter]

Thank you for sending your work entitled “A structural model of the active ribosome-bound membrane protein insertase YidC” for consideration at *eLife.* Your article has been favorably evaluated by Randy Schekman (Senior editor) and 2 reviewers, one of whom is a member of our Board of Reviewing Editors.

The Reviewing editor and the other reviewers discussed their comments before we reached this decision, and the Reviewing editor has assembled the following comments to help you prepare a revised submission.

This excellent contribution by Beckmann and co-workers develops a structural model of the active ribosome-bound YidC. They derive a structural model of the YidC monomer using co-variation analysis and molecular dynamics. The Beckmann YidC structural model is very good (with a root mean square deviation between the TM helices of their model of 7.5 Angstroms) when compared to the structure of the *Bacillus halodurans* YidC2 recently published in Nature. Both the *B. h* structure and Beckmann model possess a hydrophilic groove within the membrane embedded region that is open to the cytoplasm and lipid bilayer. However, the Beckmann model is less open (which makes sense) because they imposed TM3/TM5 interactions at Pro425-Pro499 and TM3/TM6 interactions at both Cys423-Gln528 and Phe 433-Thr524. It may represent a different conformational state of YidC than seen in the *B. h* structure, and this could reflect some of the differences in the structures.

The YidC model was then used to reconstruct the structural features of a translating YidC-ribosome complex with a bound subunit c of ATP synthase (F_o_c), determined by cryo-electron microscopy. The reconstruction model is quite good. Of the two unknown membrane densities in their model, one was suggested to be TM1 of the *E. coli* YidC (not present in their model) and the other was the hydrophobic TM segment of F_o_c. Overall, this work presenting a structural model of the active ribosome-bound F_o_C YidC complex will have a significant impact within the YidC field and the membrane field in general.

The main essential point for improvement agreed by both referees is the rigorous assignment of TM1 and the TM segment of F_o_c in their structure. This is an important part of this paper, and is worth nailing down. The crosslinking experiment as presented is incomplete as there is no suitable negative control (i.e., a cysteine position that does not crosslink to substrate). In short, the authors have two extra unaccounted densities (near helix 3 and near helix 5). At a minimum, the authors should place a cysteine in either helix 3 or helix 5, and directly compare substrate crosslinks. As currently depicted, one cannot evaluate the specificity of the crosslink and therefore the validity of the assignment of the nascent chain helix in their structure.

---

## [Author Response]

*[…] The main essential point for improvement agreed by both referees is the rigorous assignment of TM1 and the TM segment of F*_*o*_*c in their structure. This is an important part of this paper, and is worth nailing down. The crosslinking experiment as presented is incomplete as there is no suitable negative control (i.e., a cysteine position that does not crosslink to substrate). In short, the authors have two extra unaccounted densities (near helix 3 and near helix 5). At a minimum, the authors should place a cysteine in either helix 3 or helix 5, and directly compare substrate crosslinks. As currently depicted, one cannot evaluate the specificity of the crosslink and therefore the validity of the assignment of the nascent chain helix in their structure*.

As suggested by the referees we have performed additional crosslinking experiments, the results of which are now shown in Figure 3. Specifically, we have attempted to crosslink F_O_c(G23C)-RNCs to single cysteine mutants of YidC at positions P431 (one residue after the previously shown M430 in TM3), and positions V500 and T503 in TM5. In our YidC model, the latter two positions point away from the electron density that we assigned to the nascent chain, and face the electron density that we assigned to YidC-TM1. Hence, crosslinks to these positions would be expected in case our assignment of the two electron densities would be inverted.

As a result, in full agreement with our interpretation in the initial submission, YidC mutants V500C and T503C in TM5 do not crosslink to the nascent chain. An additional YidC mutant in TM3 (P431C) on the other hand does crosslink to the nascent chain, and as also expected from our model, it does so with lower efficiency than the neighboring M430C. Thus, these additional crosslinking experiments findings provide strong additional support for the correctness of our model, and we have included the results in the main text accordingly. We have moved the previous version of panel 3c, which contains the negative control with a cystein-less RNC, to Figure 3—figure supplement 2.

Taken together, these crosslink experiments indeed validate the assignment of the nascent chain helix in our structure, as requested.